# Assessment of a Loop-Mediated Isothermal Amplification (LAMP) Assay for the Rapid Detection of Pathogenic Bacteria from Respiratory Samples in Patients with Hospital-Acquired Pneumonia

**DOI:** 10.3390/microorganisms8010103

**Published:** 2020-01-11

**Authors:** Andrea Vergara, Hervé Boutal, Adrián Ceccato, Míriam López, Adrià Cruells, Leticia Bueno-Freire, Javier Moreno-Morales, Jorge Puig de la Bellacasa, Pedro Castro, Antoni Torres, Francesc Marco, Climent Casals-Pascual, Jordi Vila

**Affiliations:** 1Department of Clinical Microbiology, CDB, Hospital Clínic of Barcelona, University of Barcelona, 08036 Barcelona, Spain; VERGARA@clinic.cat (A.V.); mlopezv@clinic.cat (M.L.); jpuig@clinic.cat (J.P.d.l.B.); fmarco@clinic.cat (F.M.); ccasals@clinic.cat (C.C.-P.); 2Service de Pharmacologie et Immunoanalyse (SPI), CEA, INRA, Laboratoire d’Etudes et de Recherches en Immunonalyse, Université Paris-Saclay, F-91191 Gif-sur-Yvette, France; Herve.BOUTAL@cea.fr; 3Pulmonary and Critical Care Unit, Respiratory Institute, Hospital Clinic, 08036 Barcelona, Spain; aceccato@clinic.cat (A.C.); bueno@clinic.cat (L.B.-F.); atorres@clinic.cat (A.T.); 4Biomedical Research Centre in Respiratory Diseases (CibeRes, CB06/06/0028), Institut d’Investigacions Biomèdiques Agust Pi i Sunyer (IDIBAPS), 08036 Barcelona, Spain; 5Department of Medicine, School of Medicine, University of Barcelona, 08036 Barcelona, Spain; acruells@clinic.cat (A.C.); pcastro@clinic.cat (P.C.); 6ISGlobal, Hospital Clínic–Universitat de Barcelona, 08036 Barcelona, Spain; javier.moreno@isglobal.org; 7Medical Intensive Care Unit, Hospital Clínic, 08036 Barcelona, Spain

**Keywords:** loop-mediated isothermal amplification (LAMP), hospital-acquired pneumonia (HAP), diagnostic techniques, respiratory system, critical care

## Abstract

Rapid identification of the causative agent of hospital-acquired pneumonia (HAP) will allow an earlier administration of a more appropriate antibiotic and could improve the outcome of these patients. The aim of this study was to develop a rapid protocol to identify the main microorganisms involved in HAP by loop-mediated isothermal amplification (LAMP) directly from respiratory samples. First of all, a rapid procedure (<30 min) to extract the DNA from bronchoalveolar lavage (BAL), endotracheal aspirate (EA) or bronchoaspirate (BAS) was set up. A specific LAMP for *Staphylococcus aureus*, *Escherichia coli*, *Klebsiella pneumoniae, Pseudomonas aeruginosa*, *Stenotrophomonas maltophilia* and *Acinetobacter baumannii* was performed with the extracted solution at 65 °C for 30–40 min. Overall, 58 positive BAL and 83 EA/BAS samples were tested. The limits of detection varied according to the microorganism detected. Validation of the LAMP assay with BAL samples showed that the assay was 100% specific and 86.3% sensitive (positive predictive value of 100% and a negative predictive value of 50%) compared with culture. Meanwhile for BAS/EA samples, the assay rendered the following statistical parameters: 100% specificity, 94.6% sensitivity, 100% positive predictive value and 69.2% negative predictive value. The turnaround time including sample preparation and LAMP was circa 1 h. LAMP method may be used to detect the most frequent bacteria causing HAP. It is a simple, cheap, sensitive, specific and rapid assay.

## 1. Introduction

Hospital-acquired pneumonia (HAP) is the one that presents clinically two or more days after hospitalization and includes ventilator-associated pneumonia (VAP), which is defined as pneumonia that presents after 48 h with endotracheal intubation [1]. Patients with VAP present longer periods with mechanical ventilation, as well as longer stay in the ICU and in the hospital [2]. It is estimated that approximately 10%–40% of the patients undergoing mechanical ventilation for more than two days will develop VAP [2], with great differences among countries, type of patient, and type of intensive care unit (ICU). The implementation of different preventive strategies enabled a decrease in VAP cases, but it is still a very important problem among ventilated patients [3].

VAP is frequently caused by Gram-negative aerobic bacteria [4,5,6], such as *Pseudomonas aeruginosa*, *Klebsiella pneumoniae* and *Acinetobacter* spp; while *Staphylococcus aureus* is the most frequent Gram-positive pathogen [7]. In addition, the VAP can be polymicrobial, which complicates diagnosis and treatment. Regarding viruses, virus herpes simplex (HSV) [8] and cytomegalovirus (CMV) [9] can be reactivated in critical care patients and cause VAP. It should be noted that many of the pathogens causing VAP may present high levels of antibiotic resistance [2,7].

Unfortunately, the diagnosis of VAP is complicated and there is no reliable reference test [10]. The signs and symptoms of pneumonia, such as fever, tachycardia, leukocytosis and purulent secretions, are frequent in patients with mechanical ventilation and are not specific to VAP. Moreover, radiographic changes are difficult to interpret in this group of patients. Therefore, it is necessary to confirm the clinical suspicion of VAP through the microbiological culture of pulmonary secretions. However, the microbiological diagnosis also has several limitations: (1) difficulty to differentiate between colonization and infection; (2) non-homogeneous distribution of the infection in the lung; and (3) negative cultures due to previous empirical antibiotic treatment. Empirical treatment is usually initiated in patients with suspected VAP before having the definitive diagnosis [11], since early and adequate treatment decreases the mortality [12]. However, the indiscriminate administration of antibiotics exposes patients to unnecessary side effects, increases health care costs and favours the appearance of antibiotic resistance. Hence, the importance of knowing if there is an infection or not and the identification of the etiological agent [2].

The application of rapid diagnostic techniques to identify microbial pathogens seems to have a huge impact in the treatment of VAP, reducing inappropriate or unnecessary antimicrobial treatments and mortality in these patients [13,14]. Molecular biology techniques have allowed a faster diagnosis of VAP, especially in viral infections. They have the advantage of being faster than culture, allowing the detection of the causative agents even though the patient is receiving antibiotics and being able to quantify the bacterial concentration in the sample. On the other hand, in general they are relatively expensive techniques and detect a limited number of microorganisms.

The reaction based on the loop-mediated isothermal amplification (LAMP), developed in 2000 by Notomi et al. [15], is a rapid, simple, cheap and constant temperature nucleic acid amplification method. This method is very sensitive and specific, and there are numerous publications regarding its use in a wide range of applications [16]. LAMP is characterized by the use of 4 different primers to amplify 6 distinct regions on the target gene, an additional pair of primers to accelerate the reaction and a polymerase with high strand displacement activity in addition to a replication activity. LAMP has been used to identify respiratory pathogens, with promising results [17,18]. The purpose of this study was to develop and evaluate a rapid protocol to identify the main microorganisms involved in HAP by LAMP directly from respiratory samples.

## 2. Material and Methods

### 2.1. Study Design

We developed a rapid protocol to identify by LAMP six different bacteria (*P. aeruginosa*, *Acinetobacter baumannii*, *K. pneumoniae*, *Escherichia coli*, *S. aureus* and *Stenotrophomonas maltophilia*) and evaluated its performance compared to culture. We used three different types of samples: bronchoalveolar lavage (BAL), endotracheal aspirate (EA) and bronchoaspirate (BAS). Two different protocols were optimized and used to deal with the different consistencies of the samples processed (aspirates being more difficult to work with due to sample thickness).

### 2.2. Collection of Samples

Positive and negative BAL, EA and BAS samples were collected from the Clinical Microbiology Laboratory at the Hospital Clinic of Barcelona (Spain), after being processed for routine techniques. The microbiological result was collected, including Gram stain and culture result. BAL collected during a two-year period (2016/17) and EA/BAS over six months (January to June 2018) were stored at −80 °C until use. To standardize the protocol, negative samples were mixed to obtain a homogeneous matrix. Aliquots of this homogeneous negative sample was spiked with different microorganisms and used to determine the limits of detection for each microorganism.

### 2.3. Routine Microbiological Methods

Respiratory samples were collected in sterile containers and transported to the laboratory in less than two hours. Gram staining of the samples in the area of maximal purulence was examined for leukocytes and epithelial cells. Only respiratory samples with Murray–Washington classification degrees of IV (10–25 epithelial cells and >25 leukocytes per field using a low magnification lens (×100)), V (≤10 epithelial cells and >25 leukocytes) or VI (≤10 epithelial cells and ≤10 leukocytes) were processed for culture. Specimens not fulfilling these criteria were not considered to be representative of distal airways and were not processed for culture.

Good-quality respiratory specimens were quantitatively plated on blood and chocolate agar. Isolated bacteria were identified by matrix-assisted laser desorption/ionization time-of-flight mass spectrometry (MALDI-TOF MS) (Bruker Daltonics, Bremen, Germany). Susceptibility testing was performed according to EUCAST guidelines (Available online: www.eucast.org). Potential pathogenic bacteria included *Haemophilus influenzae*, *Streptococcus pneumoniae*, *Moraxella catarrhalis*, Gram-negative bacilli, *P. aeruginosa* and *S. aureus*. Non-potential pathogenic microorganisms included Viridans group streptococci, *Neisseria* spp, *Candida* spp, *Corynebacterium* spp, *H. parainfluenzae* and coagulase-negative staphylococci and were not considered as clinically significant. These samples were informed as having normal or mixed flora.

### 2.4. Extraction of DNA

Two protocols depending on the type of sample were performed, and they are described in Figure 1. Briefly, samples were concentrated by centrifugation and boiled in DireCtQuant 100W buffer (FrontexBioMed, Ltd./DireCtQuant, Lleida, Spain) to obtain extracted DNA. This extract was later on used for the LAMP test.

### 2.5. LAMP Protocol

Previously described primers [17] were used for *P. aeruginosa*, *A. baumannii*, *K. pneumoniae*, *E. coli*, *S. maltophilia*, and *S. aureus*. The LAMP was performed in 25 μL of reaction mixture: 5 μL primers (0.2 μM outer, 1.6 μM inner and 0.4 μM loop primers), 15 μL Isothermal Master Mix (Optigene) and 5.0 μL of extraction product. Once the reaction mix is ready, gentle vortex and centrifugation must be performed. The reaction was conducted in a Versant kPCR (Siemens, Beersel, Belgium) at 65 °C for 40 min for BAL samples and in a Lightcycler® 480 System (Roche, Sant Cugat del Vallès, Spain) at 65 °C for 30 min for EA/BAS. Both protocols can be implemented independently of the thermocycler, but we performed them in two different machines because they were done in different times and changes in laboratory equipment occurred. 

### 2.6. Determination of the Limits of Detection

Each microorganism was inoculated into negative BAL samples to a final concentration ranging from 10^7^ to 10^2^ CFU/mL. For EA/BAS it was not possible to perform the same study for sensitivity due to the consistency of the samples and the physical impossibility of obtaining homogeneous matrix for all the dilutions. Instead, positive samples were used to perform serial 10-fold dilutions in saline buffer. Final concentration for each dilution was based on the approximate initial concentration determined by culture. 

### 2.7. Statistical Analysis

Concordance between culture and LAMP results was studied. Major errors were defined as result discrepancies where the microorganism identified by LAMP was completely different from that identified in culture or the detection of a pathogen when the culture had none. Minor errors were defined as concordant results for the major pathogen identification, but LAMP identified additional microorganisms.

Diagnostic performance was based on sensitivity, specificity, negative- and positive-predictive values, and accuracy defined as described elsewhere [19,20]. Accuracy was calculated as concordant results over total number analyzed. Cohen’s kappa coefficient was also calculated. Statistical analyses were performed using Stata version 15 (College Station, Texas, USA).

### 2.8. Ethical Aspects

The Ethical Committee of the Hospital Clinic of Barcelona, Spain, approved the study protocol on 9, August, 2016. The study is registered as HCB/2016/0294.

## 3. Results

### 3.1. BAL Samples

The limit of detection of LAMP for the different pathogens tested in BAL was 10^2^ CFU/mL for *S. aureus*, *E. coli*, *P. aeruginosa* and *K. pneumoniae*, and 10^4^ CFU/mL for *S. maltophilia* and *A. baumannii*. No cross-reaction was identified using these primers while testing spiked samples with the other microorganisms included in the study. Regarding the evaluation with clinical samples, the concordance between culture and LAMP results is shown in Table 1. A total of 58 positive BAL samples were tested. All major errors happened due to a low concentration of bacterial in the sample, except for one case in which *A. baumannii* was not detected although with high concentration, and even after repeating the technique. The accuracy of the LAMP assay was of 77.6% or 87.9% if the minor errors were considered as a non-true or true result, respectively (Table 2). In addition, calculation of the clinical sensitivity and specificity yielded 100% (95%CI; 59% to 100%) specificity, a sensitivity of 86.3% (95%CI; 73.7% to 94.3%), a PPV of 100% and a NPV of 50% (95%CI; 33.5% to 66.6%) when minor errors where treated as a true positive sample. The statistical data considering minor errors as a false positive sample are shown in Table 2.

### 3.2. BAS/EA Samples

The estimated limit of detection of the LAMP assay to detect pathogens directly from BAS or EA samples was 10^2^ CFU/mL for *P. aeruginosa, K. pneumoniae* and *A. baumannii,* 10^3^ CFU/mL for *E. coli*, and 10^4^ CFU/mL for *S. aureus* and *S. maltophilia*. No cross-reaction was identified using these primers while testing spiked samples with the other microorganisms included in the study. Regarding the evaluation with clinical samples, a total of 83 samples were tested. The accuracy between culture and LAMP results is shown in Table 2 and it was of 95.2% or 72.3%, respectively when the minor errors where considered as true positives or not. In Table 3, the concordance between the LAMP assay and bacterial culture is shown. All major errors happened due to a low concentration of bacterial in the sample, except for two cases: *S. aureus* with more than 10,000 CFU/mL in culture and *A. baumannii* with 300,000 CFU/mL in culture.

Regarding possible amplification inhibitors, a mix of lipidic, hematic, viscous and mucous samples were used and inoculated with the different microorganisms. No evidence of interference was observed in any of these prepared samples.

Most positive results appeared before 30 min, and together with the sample handling time, the results could be obtained within one-hour time.

The sensitivity, specificity, negative and positive predictive values of the LAMP assay to detect pathogens directly from BAS/EA samples are shown in Table 2. When minor errors were considered as false positive results, the sensitivity was 68.9% (95%CI; 57.1% to 79.2%), the specificity was 100% (95%CI; 66.4% to 100%), and the PPV and NPV were of 100% and 28.1%, respectively. However, when minor errors were considered as true positive the statistical indicators were: 94.6% (95%CI; 86.7% to 98.5%) sensitivity, 100% (95%CI; 66.4% to 100%) specificity, 100% PPV and 69.2% (95%CI; 46.5% to 85.4%) NPV (Table 2).

The time to positivity of the LAMP test was strongly correlated with the number of bacterial CFU/mL in culture in EA/BAS samples (*r* = −0.71, *p* < 0.01) but not in BAL samples (Figure 2). These results suggest that, at least in EA/BAS samples, the time to positivity of LAMP could be used as a semi-quantitative measure.

## 4. Discussion

Although new preventive measures have led to a reduction of HAP incidence, it remains associated with important morbi-mortality [21]. Therefore, it is necessary to introduce new methods to improve an early diagnosis. Here, we have evaluated the use of LAMP as a rapid diagnostic tool to identify the main pathogens involved in HAP with promising results.

In this study, we have performed a proof of concept of the used of LAMP to detect the main pathogens causing HAP directly from clinical respiratory samples. We found an overall accuracy between LAMP and culture of 88% for BAL samples and 95% for BAS/EA samples. Almost all syndromic tests available to identify pathogens causing respiratory tract infections include virus and atypical bacteria [22], but not many include the major bacterial pathogens causing HAP. Furthermore, rapid identification of the pathogen causing HAP is crucial to improve the patient outcome. In this sense, a multiplex PCR-based syndromic panel including 16 bacterial and one fungal target as well as 22 antimicrobial resistance markers has been evaluated [23,24,25], showing a sensitivity from 60% to 90% and a turnaround time of circa 4 h. In our case, as LAMP is much less affected than PCR by possible inhibitors in the sample [26], in the condition of high bacteria concentration, no automatic nucleic acid extraction is required, and therefore we can reduce turnaround including preparation of the sample and LAMP to around 1 h.

Kang et al. [17] performed a nationwide study in China in which the qLAMP assay was compared with culture to detect eight respiratory bacterial pathogens from sputum, detecting pathogens in 1047 (69.28%) patients from 1533 qualified patients. They used a kit for the nucleic acid extraction. In our study, although the number of samples included is limited, we have demonstrated that a LAMP applied to respiratory sample without a strict extraction step is possible.

Cost is also important. We have calculated that the LAMP assay (counting both reagents for nucleic acid extraction and LAMP together with plastic material) to detect the six pathogens causing HAP in this study will cost EUR 12.

LAMP may be useful then as a complementary tool to culture, allowing rapid identification of the microorganism causing the infection. In addition, a semi-quantitative approach can be used extrapolating the time to result with the CFU/mL in BAS and EA. In BAL, the data that we had were not sufficient to get a significant result (data not shown). If the quantification is under 10²–10³ CFU/mL, the LAMP will probably not provide a positive result, but in general the culture result will be considered as colonization in these cases. Therefore, every positive LAMP result should be taken into consideration, always taking into account the Gram stain, quality of samples and clinical situation of the patient. Culture should not be avoided in any case and LAMP could be implemented as a complement to accelerate the diagnosis of HAP.

Furthermore, it also seems promising for the detection of resistance genes [27,28]. Pathogen identification and potential antibiotic resistance is possible with LAMP, both more rapidly identified (1 h) than with a time consuming (16 to 24 h) classical phenotypic method. Furthermore, it could be even faster when applied directly to samples [29].

In conclusion, the LAMP method may be used to detect the most frequent bacteria causing HAP. It is a simple, cheap, sensitive, specific, and rapid (circa one hour) assay. Multiplexing targets may facilitate the implementation of this technique in routine laboratories. Each laboratory must adapt the targets according to its epidemiology.

## Figures and Tables

**Figure 1 microorganisms-08-00103-f001:**
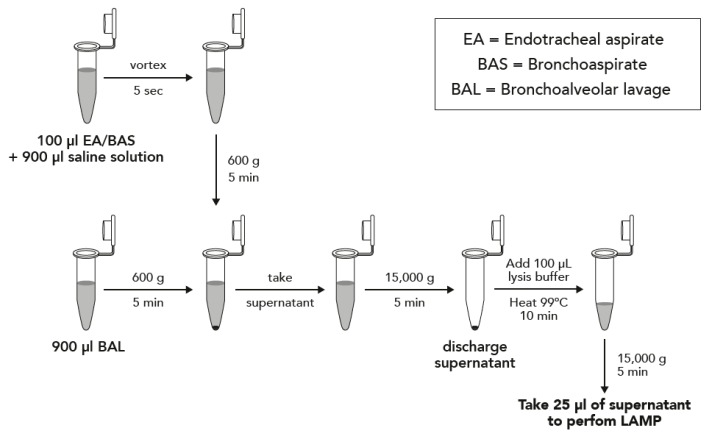
Sample preparation workflow for bronchoalveolar lavage (BAL) and bronchoaspirate/endotracheal aspirate (BAS/EA).

**Figure 2 microorganisms-08-00103-f002:**
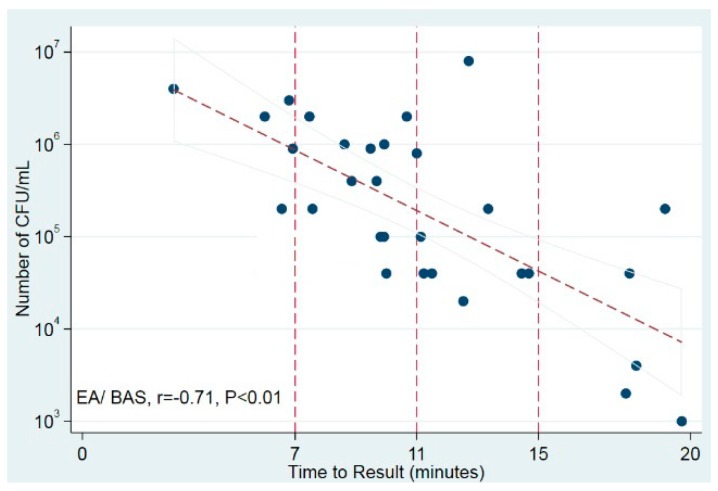
Correlation of time to positivity and bacterial load in culture. Scatter plot shows time to result (in minutes) against number of CFU/mL (log10). Trend line displays best fit of all data points and the 95% CI for the prediction. Vertical reference lines indicate time points. BAS/EA: bronchoaspirate/endotracheal aspirate.

**Table 1 microorganisms-08-00103-t001:** Loop-mediated isothermal amplification (LAMP) results from BAL samples according to the result obtained by culture.

Microorganisms Identified by Culture	N	Concordance	Minor Errors	Major Errors	Comments
*S. aureus*	19	15	2 ^A,B^	2 ^C,D^	^A^ LAMP: SAUR and SMAL detected ^B^ LAMP: SAUR and PAER detected (GNB in gram stain) ^C,D^ Culture: Few CFU of SAUR
*P. aeruginosa*	11	9	-	2 ^E,F^	^E^ Culture: < 1000 CFU of PAER, LAMP: SAUR detected ^F^ Culture: <1000 CFU of PAER, LAMP negative
*S. maltophilia*	6	6	-	-	-
*K. pneumoniae*	4	2	-	2^G^	^G^ Culture: <1000 CFU of KPNE (two cases)
*E. coli*	3	3	-	-	
*A. baumannii*	2	1	-	1 ^H^	^H^ Culture: 100,000 CFU of ABAU
Negative	7	6	1 ^I^	-	^I^ LAMP: KPNE detected (GNB in gram stain)
Mixed flora	6	3	3 ^J,K,L^	-	^J^ LAMP:PAER detected ^K^ LAMP:SAUR detected ^L^ LAMP:KPNE detected
TOTAL	58	45	6	7	

N: number of samples tested by culture and LAMP; SAUR: *S. aureus*; SMAL: *S. maltophilia;* PAER: *P. aeruginosa*; KPNE: *K pneumoniae*; ECOL: *E. coli;* ABAU*: A. baumannii;* GNB: Gram-negative bacilli.

**Table 2 microorganisms-08-00103-t002:** Statistics of the use of LAMP to detect pathogens directly from BAL and BAS/EA samples.

	Minor Error as Wrong		Minor Error as Right	
	Value (%)	95%CI	Value (%)	95%CI
BAL				
Sensitivity	76.5	62.5–87.2	86.3	73.7–94.3
Specificity	85.7	42.1–99.6	100	59–100
PPV	97.5	86.3–99.6	100	89–100
NPV	33.3	21.9–47.2	50	33.5–66.6
Accuracy	77.6	64.7–87.5	87.9	76.7–95
Kappa coefficient	43.3	17.9–68.8	70.9	51.5–90.2
BAS/EA				
Sensitivity	68.9	57.1–79.2	94.6	86.7–98.5
Specificity	100	66.3–100	100	66.4–100
PPV	100	91.3–100	100	93.5–100
NPV	28.1	21.8–35.5	69.2	46.5–85.4
Accuracy	72.3	61.4–81.6	95.2	88.1–98.7
Kappa coefficient	46.3	29.2–63.5	89.7	80–99.5

BAL: bronchoalveolar lavage; BAS: bronchoaspirate; EA: endotraqueal aspirate; PPV: positive predictive value; NPV: negative predictive value; CI: confidence interval.

**Table 3 microorganisms-08-00103-t003:** LAMP results from BAS/EA samples according to the result obtained by culture.

Microorganisms Identified by Culture	N	Concordance	Minor Errors	Major Errors	Comments
*K. pneumoniae*	13	10	3 ^A,B^	-	^A^ LAMP: KPNE and ECOL detected (two cases) ^B^ LAMP: KPNE and PAER detected
*S. aureus*	10	7	2 ^C,D^	1 ^E^	^C^ LAMP: SAUR and KPNE detected ^D^ LAMP: SAUR and ECOL detected ^E^Culture: <1000 CFU of SAUR
*P. aeruginosa*	11	9	2 ^F^	-	^F^ LAMP: PAER and ECOL detected (two cases)
*E. coli*	8	3	5 ^G,H,I,J^	-	^G^ LAMP: ECOL and PAER detected ^H^ LAMP: ECOL, PAER, SAUR and SMAL detected ^I^ LAMP: ECOL and KPNE detected ^J^ LAMP: ECOL, PAER and SMAL detected (two cases)
*S. maltophilia*	2	2	-	-	-
Polymicrobial	10	3	4	3	See Appendix A.
Other	10	7	3 ^K,L,M^	-	^K^ LAMP: ECOL and KPNE detected ^L^ LAMP: PAER detected ^M^ LAMP: PAER, SAUR and SMAL detected
Mixed flora	10	10	-	-	-
Negative	9	9	-	-	-
TOTAL	83	60	19	4	

N: number of samples tested by culture and LAMP; KPNE: *K pneumoniae*; ECOL: *E. coli*; PAER: *P. aeruginosa*; SAUR: *S. aureus*; SMAL: *S. maltophilia*.

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
