# Peer review of "Assessment of a Loop-Mediated Isothermal Amplification (LAMP) Assay for the Rapid Detection of Pathogenic Bacteria from Respiratory Samples in Patients with Hospital-Acquired Pneumonia"

_microorganisms, 2020, doi:10.3390/microorganisms8010103_

Round 1
Reviewer 1 Report
strength : a novel study
limitation: case number
however , this draft have merit for publication
I have some comments
1 . the HAP including the non-tuberculosis mycobacterium, the LAMP can detect the NTM in the speciemens or differentiate the BACTERIA from the NTM?
2 how to classify the mexed flora?or definition of the mixed flora and polymicrobial?
3 the sensivity of the BAS/EA is too low?
4 the number of the study patients is small , the conclusion may be more conservative?
5 the LAMP may have the brief discussion
Author Response
Thank you very much for your comments.
Reviewer 1
1. The HAP including the non-tuberculosis mycobacterium, the LAMP can detect the NTM in the speciemens or differentiate the BACTERIA from the NTM?
Although it was not the objective of this study since we focused in the main pathogens causing HAP (and in particular VAP), a LAMP to differentiate NTM could be designed if necessary.
2. How to classify the mexed flora?or definition of the mixed flora and polymicrobial?
Mixed flora is considered when more than one microorganism is isolated/detected in a clinical specimen but they are not considered clinically significant. Conversely, polymicrobial indicates that more than one microorganism is found in the same sample and they are considered as clinically significant.
3. The sensivity of the BAS/EA is too low?
The sensitivity was 94.6% if we consider minor errors as true positives. As described in the Material and Methods Section (page 4), minor errors were defined as concordant results for the major pathogen identification but with an additional microorganism identified by LAMP (Table 2, pag.6).
4. The number of the study patients is small, the conclusion may be more conservative?
In this study 58 BAL samples and 83 BAS/EA were included. Although the study sample size had sufficient statistical power to find significant differences we concur with the reviewer that further validation studies will be required to provide further proof on how these results can be used clinically (lines 222-223). The following sentence has been added into the discussion section (lines 237-238): “In our study, although the number of samples included is limited, we have demonstrated that a LAMP applied to respiratory sample without a strict extraction step is possible.”
5. The LAMP may have the brief discussion.
A brief description of this methodology has been added into the Introduction Section (lines 82-84): “LAMP is characterized by the use of 4 different primers to amplify 6 distinct regions on the target gene, an additional pair of primers to accelerate the reaction and a polymerase with high strand displacement activity in addition to a replication activity.”
Reviewer 2 Report
The manuscript described the current situation and future solutions of hospital-acquired pneumonia (HAP) such as ventilator-associated pneumonia (VAP). VAP, caused by Gram-negative aerobic bacteria, is a serious problem in hospitals. Thus, preventive strategies have to be carried out. However, VAP diagnosis is complicated. The authors developed the method to identify the main microorganisms involved in HAP by LAMP directly from respiratory samples. This is expected to be a promising method to diagnose HAP. Therefore, the manuscript is not too excellent to be published. In other words, the manuscript is so excellent that it should be published. Comments (1) (in line of 58) Is it easy to distinguish VAP from other types of pneumonia? (2) How much are LAMP results from samples reliable? Accuracies are 88% for BAL samples and 95% for BAS/EA samples, aren’t they? (3) Do pathogens different from the six pathogens described here cause VAP? That is all.
Author Response
Thank you very much for your comments.
Reviewer 2
(1) (in line of 58) Is it easy to distinguish VAP from other types of pneumonia?
VAP is the pneumonia that presents after 48 hours with endotracheal intubation (line 43-45). The signs and symptoms of pneumonia are not specific for VAP (line 59), but common to other conditions and, of course, other types of pneumonia.
(2) How much are LAMP results from samples reliable? Accuracies are 88% for BAL samples and 95% for BAS/EA samples, aren’t they?
Accuracies are 88% for BAL and 95% for BAS/EA samples when minor errors were considered as true positives. This allows us to obtain a rapid and reliable result, taking into account that the bacterial culture will be performed in parallel.
(3) Do pathogens different from the six pathogens described here cause VAP?
In our setting, the pathogens selected are the bacteria most frequently identified as causative agents of VAP. However, other Enterobacteriaceae, such as Proteus or Serratia, are less frequently involved as causative agents of VAP. Importantly, LAMP allows for the inclusion of other microorganisms and can be easily adapted to the local epidemiology (Lines 257-258).